# The Automatization of a New Thermography Method Using Invasive Nociceptive Stimulation to Confirm an Autonomic Phenomenon within a Trigger Point Referred Pain Zone

**DOI:** 10.3390/brainsci11070893

**Published:** 2021-07-06

**Authors:** Elżbieta Skorupska, Tomasz Dybek, Michał Rychlik, Marta Jokiel, Paweł Dobrakowski

**Affiliations:** 1Department of Physiotherapy, Poznan University of Medical Sciences, 61-701 Poznań, Poland; marta.jokiel@gmail.com; 2Department of Physiotherapy, Opole University of Technology, 45-758 Opole, Poland; dtdybek@gmail.com; 3Department of Virtual Engineering, Poznan University of Technology, 60-965 Poznań, Poland; rychlik.michal@poczta.fm; 4Department of Traumatology, Orthopedics and Hand Surgery, Poznan University of Medical Sciences, 61-701 Poznań, Poland; 5Psychology Institute, Humanitas University in Sosnowiec, 41-200 Sosnowiec, Poland; paweldobrakowski@interia.pl

**Keywords:** active dynamic thermography, low back pain, sciatica, muscle pain, MATLAB

## Abstract

The trigger points (TrPs) related to chronic low back pain that mimic sciatica have been lately recognized and included in the International Classification of Diseases, 11th Revision. This study examined the MATLAB software utility for the objective stratification of low back pain patients using the Minimally Invasive Procedure (MIP). The two diagnostic MIP parameters were: average temperature (ΔTavr) and autonomic referred pain (AURP). Chronic sciatica patients with TrPs (*n* = 20) and without TrPs (*n* = 20) were examined using the MIP. A significant increase in both parameters was confirmed for the thigh ROI of the TrP-positive patients, with ΔTavr being the leading parameter (*p* = 0.016, Exp(β) = 2.603). A continued significance of both parameters was confirmed from 6′00″ to 15′30″ (*p* < 0.05). The maximum AURP value was confirmed at 13′30″ (*p* < 0.05) (TrPs(+) 20.4 ± 19.9% vs. TrPs(-) 3.77 ± 9.14%; *p* = 0.000; CI (0.347,0.348)).

## 1. Introduction

One of the technologies applied in modern medicine is imaging performed with an infrared thermography (IRT) camera [1]. The examination of cutaneous IRT assumes the identification of: (i) significant averaged and asymmetric temperature parameters; (ii) thermoregulatory malfunctions/alternations related to some functional perturbations of the autonomic nervous system as the mark of a disease presence; (iii) physiological information to monitor blood flow, cardiac pulse, etc.; and (iv) alternations in the main thermoregulatory functions [1]. The biological background for IRT measurement is a complex reaction between the blood flow rate and the local structures of subcutaneous tissues under the regulation of the sympathetic nervous system. The skin temperature in healthy individuals is symmetric [2]. Thus, a thermal symmetry assessment is considered a valuable method of evaluating physiological normality/abnormality.

Two types of thermography are used in medicine: static thermography (ST) and active dynamic thermography (ADT). ST is a qualitative method of a nonobjective visual analysis of a single image mainly based on the confirmation of some asymmetric temperature changes in the pathological region. Thermal asymmetries greater than 0.5–0.7 °C are usually associated with a dysfunction of the musculoskeletal system [2]. ADT is a modern quantitative diagnostic method using external thermal stimuli, exercise, or pressure excitation to provoke a transient, amplified autonomic nervous system (ANS) response. It provides a more visible image contrast and allows a more precise analysis of pathological skin microcirculation [3]. ADT protocols are highly recommended to support the diagnosis of breast tumor, Raynaud’s disease, burn wounds, or pain states [1]. In pain medicine, the ADT utility has been indicated for neural diseases, musculoskeletal diseases, inflammatory diseases, and vascular diseases. High IRT reliability has also been confirmed for muscle examination [4,5,6]. Recently, an increasing interest in using an infrared thermography camera as a tool for an objective confirmation of trigger points (TrPs) has been observed [7,8,9]. Trigger points are characteristic of regional myofascial pain syndrome, which is a very common problem that accompanies multiple medical conditions and negatively influences the life quality of affected patients [10,11]. A trigger point is defined as “a hyperirritable spot within a taut band of a skeletal muscle that is painful on compression, stretch, overload, or contraction of the tissue which usually responds with a referred pain that is perceived distant from the spot” [12]. The main problem arousing controversy is the lack of a gold standard that could be used to objectively confirm trigger points. Currently, the palpatory diagnostic criteria established decades ago by Travell and Simons are still applied to TrPs diagnosis [13].

The Minimally Invasive Procedure (MIP) is a new type of ADT proposed to confirm autonomic phenomenon observed within the referred pain zone due to the noxious stimulus of a trigger point. Very likely, this new proposal may be a valuable tool for an objective TrP confirmation [14].

A series of papers showing amplified vasodilatation reactivity confirmed by the MIP within the gluteus minimus referred pain zone of sciatica patients, has been published [15,16,17]. It seems very useful, especially because active trigger points within the gluteus minimus muscle have been considered the main source of gluteal syndrome that mimics sciatica and that has been included in the 11th Revision of the International Classification of Diseases (ICD-11) for the first time [18,19]. Thus, a further development of the MIP seems particularly important for a possible objective trigger points confirmation. The main idea behind this method is the nociceptive stimulation of trigger points to provoke an amplified vasomotor response following the referred pain pattern (the patient’s daily complaint) [17]. Two parameters of the MIP, namely the autonomic referred pain (AURP) occurrence and a significant increase in the average temperature (Tavr), were identified as diagnostic parameters indicative of a trigger point presence. The AURP reflected “a high or low temperature mark on the pain region” limited to the pain perceived during the MIP and coincident with both the patient’s daily complaint and the referred pain pattern defined by Travell and Simons [17]. This parameter demanded thermogram segmentation, and it is widely known that the accuracy of a diagnosis depends on how well the segmentation of the region of interest (ROI) can be performed [20]. The quality of the MIP interpretation can be improved thanks to using advanced thermogram software. The manual thermal data analysis was the main disadvantage of the MIP. The calculations were limited to a few thermograms representative of each step of the procedure, consisting of 350 thermograms overall. Moreover, the manual thermal data analysis was highly time-consuming and slowed down the rate of the studies considerably. Due to the manual data analysis application, some crucial information about the nature of the two diagnostic parameters of the MIP can still be overlooked. The available thermography processing software dedicated to medicine is not applicable to the MIP. The analysis of the MIP results demands specific data segmentation that can show the temperature changes in the ROI during the entire stimulation and observation phases that last 16 min total. Developing new software dedicated to MIP is necessary. First, however, an automated data analysis should be performed to define the nature of the AURP and the average temperature changes during the whole procedure step by step. The automated analysis of the series of thermograms obtained for the groups of patients can be performed using automated data analysis software such as MATLAB, which has been used for thermogram analyses before [21]. However, for MATLAB to be able to confirm the autonomic phenomenon within the trigger point referred pain zone, some new algorithms of the MIP data analysis need to be applied.

The aim of the study was to assess the utility of MATLAB software for a more precise description of the trigger point-related autonomic phenomenon confirmed by the Minimally Invasive Procedure.

## 2. Materials and Methods

This study was conducted in accordance with the Declaration of Helsinki and was approved by Ethics Committee of Poznan University of Medical Sciences no. 689/20. All subjects gave written informed consent to participate in the study before data collection. A detailed description of all examination and treatment procedures, including the Minimally Invasive Procedure, as well as of the risks involved in the study was provided to the participants. The participants had the right to refuse to undergo the Minimally Invasive Procedure and withdraw from the study at any time without penalty.

### 2.1. Participant Characteristics

Forty chronic sciatica subjects (mean age 40.2 + 5.3) with pain symptoms lasting for more than 6 months or in the recurrent state were recruited from the University Pain Clinic and enrolled in the study. Twenty sciatica subjects (*n* = 20) presenting with active gluteus minimus trigger points (TrPs) comorbidity and twenty chronic Trp-negative sciatica patients (*n* = 20) participated in the study. There was no significant difference between the Trp-positive and Trp-negative sciatica patients. The demographic data were, respectively, as follows: (i) age 44 ± 6.6 vs. 46 ± 7.5, (ii) pain duration 11.3 ± 7.7 vs. 10.7 ± 7.5, (iii) VAS 6.17 ± 2,1 vs. 5.35 ± 1.6, and (iv) BMI 23.29 ± 3.1 vs. 26.49 ± 4.1. The diagnosis of the TrP presence within the gluteus minimus muscle was based in the current study on Travell and Simons’ clinical criteria [13]. However, due to the lack of the taut band of the gluteus minimus muscle, the presence of the referred pain pattern had to be confirmed for the TrP diagnosis. Active trigger points within the gluteus minimus were confirmed if spot tenderness, pain recognition, and a limited range of movement were present together with the full referred pain pattern revealed in the thigh with/without the calf (after it was evoked by snapping palpation). All subjects were re-diagnosed by an independent University neurologist, and the diagnosis of sciatica was confirmed based on the positive straight leg raise test, bedside examination, and magnetic resonance imaging. Subjects were included if they were diagnosed with sciatica and if they met the following inclusion criteria: age between 30 and 60 (inclusive), both lower limbs present, pain duration of 1–6 months, and >3 on the 1–10 point visual analog scale (VAS) of leg pain, with this being the dominant pain problem. Subjects were excluded if they presented with: complex regional pain syndrome, cauda equina syndrome, a previous back surgery, L4-S1 root compression confirmed by magnetic resonance imaging, spinal tumors, scoliosis, pregnancy, coagulant treatment, disseminated intravascular coagulation, diabetes, epilepsy, infection, inflammatory rheumatologic diseases, stroke, or oncological history.

### 2.2. Methods

A retrospective analysis of the thermographic data extracted from previously published studies was performed [15,22]. However, the present MATLAB analysis included 95% of the unpublished thermal data (MATLAB version R2021). Our previous studies showed 3 representative thermograms out of 320 thermograms recorded during the 16-min procedure. The thermograms recorded during the MIP were grouped according to the gluteus minimus trigger points presence confirmed by Travell and Simons’ clinical criteria. Additionally, the referred pain presence was demanded for the Trp-positive diagnosis. A flow diagram of the study design is shown in Figure 1.

#### 2.2.1. Measurement Protocol

The Minimally Invasive Procedure is intended to confirm the vasomotor response within the referred pain zone of a trigger point. According to the validation study, two parameters are indicative of a TrP presence: ΔT_avr_ and AURP (both validity and reliability (test–retest). The AURP is an area of amplified vasodilatation or amplified vasoconstriction calculated as a percentage. The structure of the MIP is typical for active dynamic thermography, i.e., it includes: (1) the pre-stimulation phase—to check the thermal symmetry between both sides in order to exclude thermal asymmetry and temperature decrease of more than 0.5 °C in the pain region (a sign of neuropathic pain), (2) the noxious stimulation phase—10 min of fast—in fast–out dry needling of two trigger points/or sensitive areas (Trp-negative patients) within a muscle and the IRT observation of the region assumed to be the referred pain zone of the examined trigger point/muscle, and (3) the post-stimulation (IRT observation) phase—6 min of further IRT observation of the referred pain zone with the patient at rest. Finally, all thermograms collected during the 16-min procedure were automatically analyzed. An illustration of the MIP method is shown in Figure 2.

#### 2.2.2. Procedure in MATLAB

A general scheme of the procedure for the automated processing of thermographic images obtained during the Minimally Invasive Procedure and the analysis of all the data recorded during the examination part is shown in Figure 3. The procedure involves: (1) the manual generation of mask images described as Manual procedure A, (2) the conversion of mask images to the matrix described as MATLAB procedure B, and (3) the calculation of features and measures performed in the MATLAB environment described as MATLAB procedure C.

##### Manual Procedure A

Before the automated data analysis was performed in the MATLAB environment, images with a marked ROI (region of interest), i.e., the so-called masks, were prepared manually. The masks were defined anatomically, i.e., thigh, calf, and foot. This procedure was performed for each patient separately using paint editing software. Automatic mask imposition was impossible due to different sizes of the lower limbs in individual patients and a different position of the observed lower limb for different subjects. At this time, no procedure has been developed to automatically detect the foot, calf, and thigh in a thermographic image. As a result of the manual procedure, the so-called FOOT, CALF, and THIGH masks were developed. A sample visualization of the masks is shown in Figure 4.

##### MATLAB Procedure B

Thermograms with masks in the form of a BMP file were loaded into the MATLAB environment to convert them into matrices, which was necessary for further analyses. The procedure of thermogram processing was as follows:(1)Conversion from a true RGB color image to grayscale using the rgb2gray function. This task was performed by eliminating the hue and saturation information while retaining the luminance. The grayscale values were calculated according to the following formula: 0.299 * R + 0.587 * G + 0.114 * B, where R is the red component, G the green component, and B the blue component. As a result, a 320 × 240 matrix of values in the range from 0 to 255 uint8 was created.(2)Conversion of the uint8 values to the float format using the double function.(3)Reduction of the shadows by eliminating all values but the 255 value, which corresponded to the ROI.(4)Division of the matrix by 255, which created a “0”/”1” matrix, where “1” corresponded to the number of pixels within the ROI.

Upon the completion of thermogram processing by MATLAB, the matrices for the three ROIs were obtained as visualized in Figure 5.

##### MATLAB Procedure C

The MIP thermographic images of each patient were exported as text files. They were then loaded into the MATLAB environment and stored as a matrix sized 320 × 240. Each element of the matrix corresponded to a temperature value rounded to one decimal place. The matrices were then multiplied by mask matrices for the three areas: THIGH, CALF, and FOOT. The mask matrices were calculated according to MATLAB procedure B. The obtained matrices were saved for further calculations.

This was followed by a data-cleaning procedure, including, in particular, the deletion of patient data if the measurements took less than 15 min (due to a recording error of the thermal camera) or if the limb was significantly moved during the measurement, which made the automated calculation of the desired features and measures impossible.

##### Used Measurements and How They Were Calculated

The evaluation of the MIP diagnostic parameters included the calculation of the percentage area with a temperature response named AURP, calculated separately for vasodilatation (AURP↑) and vasoconstriction (AURP↓). The second analyzed parameter was the level of change in the mean temperature of the area (ΔT_avr_).

The procedure for the calculation of the features and measures:(1)Calculation of the area with the AURP temperature response in the ROI
(a)Calculate the values of the minimum T_min_ and maximum T_max_ temperature in the first thermogram at the moment t_0_ = 0 s.
(1)$$\vee T_{px}\in \left\{1,\;2,\;\ldots ,n\right\}\wedge \;t=t_{0},\;\;\;\;T_{max}=\max\left(T_{px1},T_{px2},\;\ldots ,\;T_{pxn}\right)$$
(2)$$\vee T_{px}\in \left\{1,\;2,\;\ldots ,n\right\}\;\wedge \;t=t_{0},\;\;\;\;T_{min}=\min\left(T_{px1},T_{px2},\;\ldots ,\;T_{pxn}\right)$$
(b)Calculate the area of the ROI surface $A_{ROI}$ for each patient as the sum of non-zero pixels in the thermogram according to the following formula:
(3)$$\vee T_{px}>0,\;\;A_{ROI}=\sum T_{px}$$where $T_{px}$—temperature value at pixel *px* within the ROI.(c)Calculate the percentage of the area with a temperature equal to T_min_$,A_{min}$and equal to T_max_, $A_{max}\;$ at the moment t_0_ according to the following formula:(4)$$\vee T_{px}\;:T_{px}=T_{max}\;\wedge \;t=t_{0},\;\;A_{max}=\;\;\frac{\sum T_{px}}{\;A_{ROI}}*100\%\;$$
(5)$$\vee T_{px}\;:T_{px}=T_{min}\;\;\wedge \;t=t_{0},\;\;A_{min}=\;\;\frac{\sum T_{px}}{\;A_{ROI}}*100\%\;$$
(d)Calculate the percentage of the area with a temperature below T_min_ + 1.5 °C for AURP↓ according to the following formula:
(6)$$\vee T_{px}\;:T_{px}\leq T_{min}+1.5\;℃,\;AURP↓=\;\;\frac{\sum T_{px}}{\;A_{ROI}}*100\%$$
(e)Calculate the percentage of the area with a temperature above T_max_-1.5 °C for AURP↑ according to the following formula:
(7)$$\vee T_{px}\;:T_{px}\geq T_{max}-1.5\;℃,\;AURP↑=\;\;\frac{\sum T_{px}}{\;A_{ROI}}*100\%$$
(2)Calculation of the change in the mean temperature of the ROI
(a)Calculate the value of the arithmetic mean temperature T_avg_tx_ in subsequent thermograms at the moment t_x_, where x = {0, 3, 6, 9, …, 900} s.(b)Calculate the value of temperature changes ΔT_avr_tx_ at the t_x_ moments as the differences between the average temperature values at successive time instants and the value at t_0_ according to the following formula: ΔT_avr_tx_ = T_avg_tx_ − T_avg_t0_.



#### 2.2.3. Statistical Analysis

Exact two-tailed Mann–Whitney *U* tests with corrected ties were performed in order to check the differences between the Trp-negative (*n* = 20) and Trp-positive (*n* = 20) sciatica patients. Tests were applied to compare the differences for AURP and ΔTavr between the aforementioned groups. Additionally, Firth’s Bias-Reduced Logistic Regression was performed to check which of the five examined components differentiate Trp-positive and TrP negative sciatica patients. If the Exp(β) was lower than 1, the expected value of the examined variable decreased and vice versa; if the Exp(β) was greater than 1, the expected value of the examined variable increased. Therefore, on the one hand, an Exp(β) of 0.8 meant that, as the independent variables increased, the chance to obtain higher values for the dependent variable was lower by 20%. On the other hand, an Exp(β) of 1.5 meant that, as the independent variables increased, the chance to obtain higher values for the dependent variable was higher by 50%. Values, figures, and tables in the text are expressed as ± standard deviation (SD). The significance level was set for all tests at *p* < 0.05. Statistical analysis was performed using IBM SPSS Statistics version 26 (IBM Corp. Released 2019. IBM SPSS Statistics for Windows, Version 26.0. Armonk, NY, USA: IBM Corp.) and MATLAB version R2021. Firth’s Regression was prepared in R version 4.0.5 using the logistf package.

## 3. Results

The MATLAB analysis trends for ΔTavr and AURP shown for each anatomical ROI (thigh, calf, and foot) of the Trp-positive and Trp-negative sciatica subjects are presented in Figure 6. Every 30 s, the data analysis of the Trp-positive and Trp-negative sciatica patients confirmed a significant difference for both (**) parameters together (Figure 1; *p* < 0.05; Mann–Whitney *U* test).

The MIP confirmed amplified vasodilatation in the thighs and calves of the Trp-positive patients. Firth’s Bias-Reduced Logistic Regression confirmed that ΔTavr is the leading diagnostic parameter to differentiate Trp-positive and Trp-negative sciatica patients (*p* = 0.016, Exp(β) = 2.603). A significant increase in both the MIP diagnostic parameters (AURP and ΔTavr) was confirmed for the TrP-positive sciatica patients in the thigh ROI (at 3′30″ of the stimulation; *p* < 0.05; Mann–Whitney *U* test). A continued significance of both parameters was confirmed from 6′00″ to 15′30″ of the procedure. The TrP-positive patients presented a significant maximum ΔTavr increase at the 16th minute of the procedure (observation phase): TrPs 0.98 ± 0.69 °C vs. non-TrPs −0.02 ± 0.83 °C (*p* = 0.028; CI (0.294,0.313)). The maximum value of the second parameter was confirmed for the Trp-positive subjects at 13′30″ (thigh AURP Trp-positive: 20.4 ± 19.9% vs. thigh AURP Trp-negative: 3.77 ± 9.14%; *p* = 0.000; CI (0.347,0.348)). The calf ROI was characterized by a significant AURP increase without a significant ΔTavr increase. The maximum AURP of the calf ROI sized 28.6 + 23.7% for the Trp-positive vs. 3.05 + 6.70% for the Trp-negative patients (*p* = 0.001; CI (0.336, 0.337) was confirmed at the 14th minute of the procedure (observation).

### 3.1. A Detailed Analysis of the Two MIP Diagnostic Parameters in Relation to the TrP Presence

#### 3.1.1. Thigh ROI

All of the sciatica patients (*n* = 40) presented with thigh pain as their daily complaint.

The Trp-positive sciatica patients (*n* = 20) demonstrated a significant increase in both diagnostic parameters for both MIP phases (stimulation phase and observation phase) (*p* > 0.05; Mann–Whitney *U* test). A significant ΔTavr increase was confirmed first at 30 s (*p* = 0.036; CI (0.111, 0.135)), then at 4.5 min into the stimulation (*p* = 0.012; CI (0.166, 0.177)). Next, a continued significant ΔTavr increase was observed from the 6th minute to the end of the procedure (*p* < 0.05; Mann–Whitney *U* test). The maximum and minimum ΔTavr changes were as follows: (1) for Trp-positive: a maximum increase of 0.98 + 0.69 °C (16′) (*p* = 0.028; CI (0.294, 0.313); Mann–Whitney *U* test) and the minimum of 0.28 + 0.26 °C (*p* = 0.036; CI (0.111, 0.135); Mann–Whitney *U* test) and (2) for Trp-negative: a maximum increase of 0.03 + 0.66 °C (14′) (not significant) and a minimum of −0.15 + 0.76 °C (15′30″) (not significant). A significant ΔTavr increase of above 0.5 °C was confirmed for the Trp-positive patients at 4′30″ of the procedure (*p* = 0.012; CI (0.165, 0.176)). Detailed average temperature changes in the thigh ROI observed every 30 s of the procedure are shown in Table 1.

Stimulation phase: The Trp-positive compared to Trp-negative patients presented a significantly higher value of AURP (*p* > 0.05; Mann–Whitney *U* test) and ΔTavr increase (*p* < 0.05; Mann–Whitney *U* test).

For the Trp-positive subjects, the maximum value of the ΔTavr increase was 0.68 + 0.69 °C at the 9th minute, and the minimum value was 0.28 + 0.26 °C at 30 s. The maximum value of the AURP size for the Trp-positive subjects was 17.9 ± 18.7% at the 9th minute, and the minimum value was 7.85 ± 8.08% at 30 s.

Observation phase: The Trp-positive compared to Trp-negative patients presented a significantly higher value of AURP (*p* > 0.05; Mann–Whitney *U* test) and ΔTavr increase (*p* < 0.05; Mann–Whitney *U* test). The maximum value of ΔTavr increase was 0.98 + 0.69 °C (16′) for the Trp-positive patients. The minimum value of ΔTavr increase for the Trp-positive subjects was 0.54 + 0.70 °C (11′). The maximum AURP size for Trp-positive subjects was 20.4 ± 19.9% (*p* = 0.000; CI (0.347,0.348)) at 13′30″, and the minimum size was 16.0% ± 17.9% at 14′30″.

#### 3.1.2. Calf ROI

Calf pain was reported by *n* = 9 of the Trp-positive and *n* = 8 of the Trp-negative sciatica patients.

There was no significant increase in both diagnostic parameters. There was no significant ΔTavr difference between the Trp-positive and -negative patients. The first significant AURP increase was confirmed at the 4th minute (19.9 + 20.2%; *p* = 0.005; CI (0.211, 0.215)). Next, a continued significant AURP increase was confirmed to last until the end of the procedure (*p* < 0.05; Mann–Whitney *U* test).

Stimulation phase: The Trp-positive patients presented a maximum AURP size of 20.8 ± 23.1% at the 9th minute (*p* = 0.002; CI (0.262,0.264)) and a minimum size of 19.9 ± 20.2% at the 4th minute (*p* = 0.046; CI (0.089, 0.130)) of the noxious stimulation.

Observation phase: The Trp-positive patients presented a maximum AURP size of 28.6 ± 23.7% at the 14th minute (*p* = 0.001; CI (0.336,0.337)) and a minimum size of 19.9 ± 23.9% at the 4th minute (*p* = 0.002; CI (0.265, 0.266)) of the noxious stimulation.

#### 3.1.3. Foot ROI

Foot pain was reported by *n* = 2 of the Trp-positive and *n* = 1 of the Trp-negative sciatica patients.

A temporary significant ΔTavr increase (at the first minute) was seen for the Trp-positive subjects (*p* < 0.05), and no significant results for both MIP diagnostic parameters were confirmed.

## 4. Discussion

This was the first time that MATLAB was used to analyze the MIP results. When compared to the manual analysis presented in our previous studies, this automatic MATLAB analysis better revealed the biological nature of the autonomic phenomenon typical of sciatica patients with gluteus minimus trigger points. Firstly, it was shown that the end of the noxious stimulation was characterized by a further ΔTavr increase and AURP development, which could not be seen when the manual analysis was applied (Figure 6). It is very interesting why vasodilatation was still growing after the noxious stimulation ends. Our hypothetical explanation is based on the fact that the muscle referred pain was considered a central phenomenon whose size grew depending on the central sensitization (CS) involvement [23]. Both the size and localization of the referred pain area can be used as a proxy for CS involved in different functional pain disorders [24]. Additionally, it proved that, if repeated stimuli are applied, the pain will increase during stimulation and the referred pain area will expand [25]. Thus, it can be speculated that this further post-noxious stimulation autonomic phenomenon (AURP and ΔTavr) increase was caused by temporal summation, which is characteristic of CS and can be measured, e.g., by repeated electrical or mechanical stimulations [26]. Additionally, an interesting MATLAB observation was that the Trp-negative subjects demonstrated small thigh AURP reactivity (Figure 6). It has been confirmed that dry needling used as a noxious stimulant can provoke temporary ANS reactivity within the referred pain zone (registered by IRT) [15]. The observed phenomenon occurs probably due to the activation of the non-noradrenergic active vasodilator system affected by the processes of reflex cutaneous vasoconstriction and vasodilatation. It is commonly known that the “activator” can be a long-term factor, e.g., illness, or a short-term factor, e.g., noxious stimulant [27,28]. Additionally, Nickel et al. [29] suggested that the ANS response observed due to noxious stimulation depends on the noxious intensity rather than the pain level. Due to the gluteus minimus anatomy and the fact that a TrP presence was confirmed by palpatory criteria, it can be assumed that some of the Trp-negative sciatica subjects presented “weak” or deeply located, not palpable TrPs, which were overlooked. For future studies, it seems interesting to analyze the needle reactivity during the MIP in sciatica patients who, according to the diagnostic palpatory criteria, were not classified as Trp-positive.

### Detailed Description of the MIP Diagnostic Parameters Using MATLAB

In the current study, the MATLAB analysis allowed us to identify some new characteristics of the amplified vasodilatation developed in the leg pain area of the Trp-positive sciatica patients (Figure 6). Firstly, the leading diagnostic value was confirmed for the ΔTavr increase (*p* = 0.016, Exp(β) = 2.603).

A detailed thigh ROI analysis showed the first significant ΔTavr increase at 30 s of the stimulation phase (*p* = 0.036; CI (0.107, 0.131); Mann–Whitney *U* test; Table 1). However, the clinically important significance of a ΔTavr increase of above 0.5 °C was observed at 4′ 30″ (*p* < 0.05). Importantly, the validation study indicated the meaning of both MIP parameters, and a significant increase in both parameters was indeed confirmed in the current study for the thigh ROI [17]. According to the literature, the size of TrPs referred pain varies between cases, which was reflected by the MIP results [30]. All patients demonstrated pain in the thigh but only some of them in the calf and/or the foot. Thus, the number of patients with calf referred pain influenced the ΔTavr but not the AURP results (Figure 6). Further, MATLAB showed the necessity of correcting the length of the MIP phases. It seems possible to shorten the noxious stimulation phase. The first significance of both diagnostic parameters was confirmed at 3′30″ (*p* < 0.05). Thus, a 10-min noxious stimulation that provokes pain seems too long. A significant increase of both diagnostic parameters was confirmed for the last 3 min of the stimulation phase and for almost the whole observation phase. Thus, for an objective TrPs confirmation, the MIP demands a simultaneous increase in both parameters. It seems that the painful stimulation phase can be shortened, but further studies considering this aspect are necessary. The relationship between the noxious input and pain perception has been extensively studied [31]. The modalities, such as noxious mechanical stimulation, have not been extensively applied but are frequently used in clinical practice [32,33]. Moreover, the lengthening of the observation phase was indicated (Figure 6). The maximum size of the amplified vasodilation was confirmed at 13′30″ of the procedure. According to active dynamic thermography protocols, the MIP method should have a precise definition of the recovery phase [34]. The analysis of both MIP parameters showed that no return to the initial phase was obtained for the current MIP protocols. Two ways are possible to improve the protocols as regards the recovery phase. One is the lengthening of the observation phase, and the other is the shortening of the stimulation phase. Both ways can probably lead to a faster recovery. The latter seems more viable, because it would shorten the time of the painful stimulation phase. Further studies considering the time demanded for both the stimulation and observation phase are necessary.

The clinical implications of the study

The MIP is intended to objectively confirm the trigger points presence. Thus, it gives an opportunity to stop controversies around their presence. However, further studies considering other muscles are necessary.

Limitation of the study

The therapist who performed dry needling was not blinded to the trigger points diagnosis results, which could provoke possible bias.

## 5. Conclusions

MATLAB allowed a more precise description of the amplified phenomenon within the trigger points referred pain zone. Both MIP parameters, with ΔTavr as the leading indicator, showed the diagnostic importance of the thigh ROI for gluteus minimus trigger points confirmation in chronic sciatica patients. For an objective TrP confirmation, the MIP demands a simultaneous increase in both parameters. The shortening of the noxious stimulation phase and the lengthening of the observation phase are suggested for further MIP development.

## Figures and Tables

**Figure 1 brainsci-11-00893-f001:**
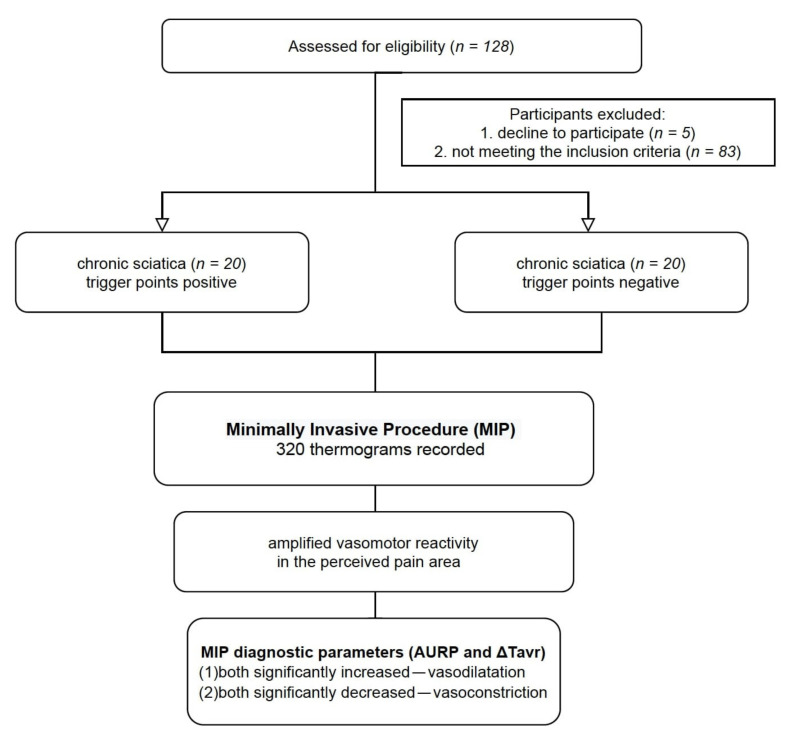
A flow diagram of the study design.

**Figure 2 brainsci-11-00893-f002:**
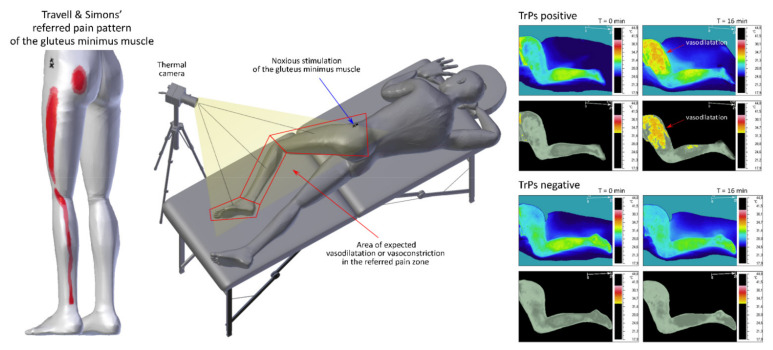
An illustration of the Minimally Invasive Procedure (MIP).

**Figure 3 brainsci-11-00893-f003:**
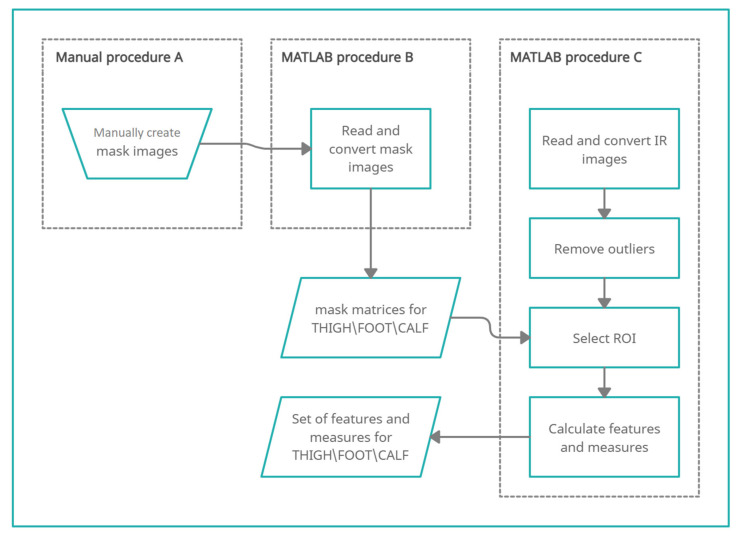
A general flow chart of the procedure for the feature and measure calculations performed in the MATLAB environment.

**Figure 4 brainsci-11-00893-f004:**
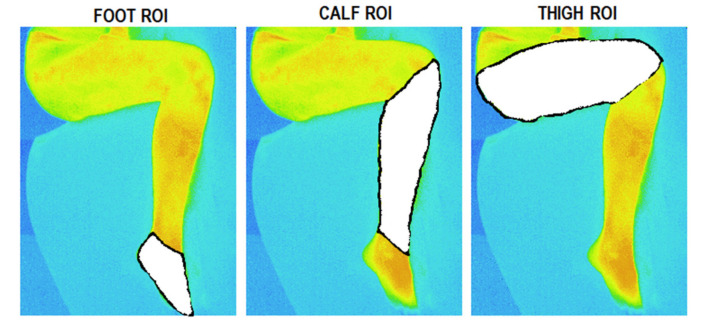
The result of manual procedure A: ROI mask creation for FOOT, CALF, and THIGH.

**Figure 5 brainsci-11-00893-f005:**
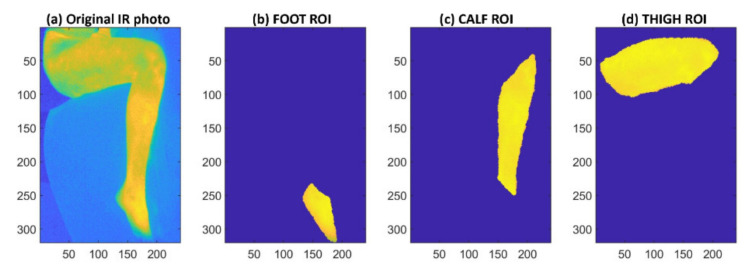
Visualization of MATLAB procedure B: The original thermogram for a random subject (**a**) and the selected ROI for FOOT (**b**), CALF (**c**), and THIGH (**d**).

**Figure 6 brainsci-11-00893-f006:**
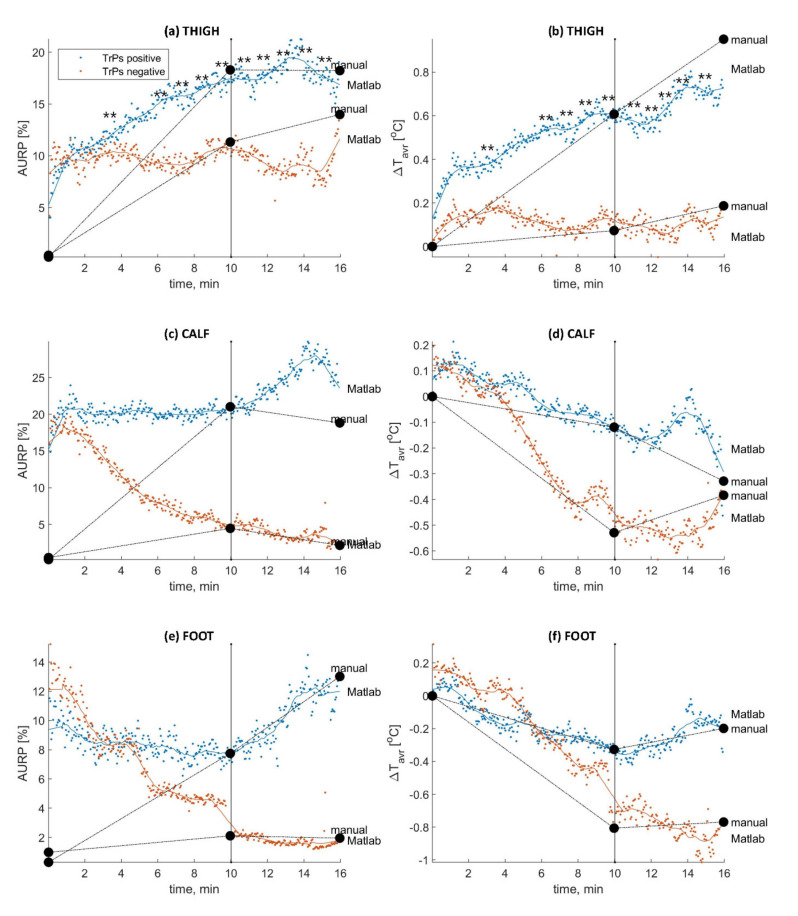
MATLAB trends for ΔTavr and AURP for each anatomical ROI (thigh (**a**,**b**), calf (**c**,**d**), and foot (**e**,**f**)) in relation to the TrP presence.

**Table 1 brainsci-11-00893-t001:** Detailed average temperature changes in the thigh ROI observed every 30 s of the procedure.

MIP Phases	Time (min)	ΔTavr		Independent Samples Mann–Whitney *U* Test	
		Trp-Positive	Trp-Negative	(CI Low, CI High)	*p*-Value
Stimulation Phase (muscle noxious stimulation)	0.5	0.27 ± 0.26	0.05 ± 0.42	0.119 (0.107,0.131)	0.036
	1	0.32 ± 0.32	0.10 ± 0.41	0.099 (0.079,0.119)	0.056
	1.5	0.37 ± 0.32	0.07 ± 0.40	0.181 (0.177,0.185)	0.009
	2	0.36 ± 0.41	0.03 ± 0.42	0.119 (0.107,0.131)	0.036
	2.5	0.32 ± 0.37	0.02 ± 0.50	0.087 (0.055,0.119)	0.074
	3	0.41 ± 0.43	0.07 ± 0.52	0.081 (0.049,0.113)	0.085
	3.5	0.38 ± 0.40	0.08 ± 0.55	0.070 (0.020,0.119)	0.116
	4	0.43 ± 0.52	0.12 ± 0.61	0.057 (−0.003,0.117)	0.158
	4.5	0.53 ± 0.45	0.04 ± 0.59	0.171 (0.165,0.176)	0.012
	5	0.49 ± 0.55	0.03 ± 0.58	0.111 (0.095,0.127)	0.046
	5.5	0.48 ± 0.57	−0.00 ± 0.62	0.158 (0.151,0.165)	0.016
	6	0.52 ± 0.56	0.00 ± 0.65	0.150 (0.142,0.157)	0.020
	6.5	0.58 ± 0.58	−0.03 ± 0.68	0.193 (0.190,0.196)	0.007
	7	0.51 ± 0.57	−0.03 ± 0.67	0.150 (0.142,0.157)	0.020
	7.5	0.55 ± 0.54	−0.03 ± 0.68	0.158 (0.151,0.165)	0.016
	8	0.59 ± 0.62	−0.04 ± 0.68	0.158 (0.151,0.165)	0.016
	8.5	0.60 ± 0.64	−0.05 ± 0.74	0.145 (0.136,0.155)	0.021
	9	0.67 ± 0.68	−0.04 ± 0.75	0.184 (0.180,0.188)	0.009
	9.5	0.60 ± 0.69	0.02 ± 0.73	0.126 (0.114,0.138)	0.033
	10	0.57 ± 0.68	−0.05 ± 0.72	0.151 (0.144,0.158)	0.021
Post-Stimulation Phase (IRT observation)	10.5	0.64 ± 0.68	−0.03 ± 0.78	0.173 (0.168,0.179)	0.013
	11	0.53 ± 0.69	−0.02 ± 0.73	0.112 (0.092,0.133)	0.051
	11.5	0.57 ± 0.75	−0.07 ± 0.74	0.161 (0.154,0.168)	0.019
	12	0.59 ± 0.74	−0.04 ± 0.76	0.133 (0.121,0.146)	0.033
	12.5	0.57 ± 0.80	−0.09 ± 0.74	0.166 (0.158,0.173)	0.017
	13	0.61 ± 0.80	−0.11 ± 0.65	0.229 (0.227,0.231)	0.004
	13.5	0.70 ± 0.74	−0.06 ± 0.73	0.215 (0.212,0.218)	0.008
	14	0.75 ± 0.78	0.03 ± 0.66	0.220 (0.216,0.224)	0.009
	14.5	0.72 ± 0.84	−0.02 ± 0.70	0.194 (0.185,0.204)	0.021
	15	0.72 ± 0.75	−0.07 ± 0.68	0.224 (0.218,0.229)	0.015
	15.5	0.68 ± 0.77	−0.14 ± 0.76	0.240 (0.230.0.249)	0.025
	16	0.97 ± 0.69	−0.02 ± 0.82	0.303 (0.294.0.313)	0.028

MIP—Minimally Invasive Procedure; ΔTavr—average temperature delta; TrPs—trigger points.

## Data Availability

The data is not publicly available due to data privacy regulations. The data presented in this study are available on request from the corresponding author.

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
