# Peer review of "The Automatization of a New Thermography Method Using Invasive Nociceptive Stimulation to Confirm an Autonomic Phenomenon within a Trigger Point Referred Pain Zone"

_brainsci, 2021, doi:10.3390/brainsci11070893_

Round 1

Reviewer 1 Report

INTRODUCTION: Despite being excellent, you should correct parts unclear in the introduction. Correcting the sentence in line 42 to 44. ST is also a quantitative and therefore objective method. As explained in the next sentence, however, correct them to greater than 0.3oC.

ST has not raised many doubts nor considered to be of limited use, there is much more literature on the reliability of ST than ADT. In ST, there is also ANS stimulation when exposing the patient to a slightly cold controlled environment at 23oC. It is the room temperature cooler than the skin that will ensure better contrast and after colors and temperature adjustments in the image (line 49) – Despite being a retrospective analysis I neither didn´t find room temperature in the method nor if the patients did a period of thermalization inside the room before MIP. And analysis accuracy depends on the instrument's calibration and good reproducibility technique (line 49).

I liked the term "autonomic referred pain (AURP)" but I do not agree in the introduction that it is only "a high temperature mark on the pain region" (line 80), there also be a decrease in temperature at the referred pain site, especially when provoked by noxious stimulus (painful compression of trigger points). So I understand that AURP has been a specific term for a temperature increase used specifically IN THIS STUDY, it cannot be generalized to the concept that there is always a temperature increase in the referred pain region. In line 152 you correctly used the expression vasodilatation or amplified VASOCONSTRICTION. Also, line 221.

METHOD

MATLAB® is a registered trademark. It should inform that it used a programming and numerical computing platform (MATLAB, MATrix LABoratory, version R2021, year).

Describe what MIP is. The reader won't know it is needling if they don't read your previous papers. It is essential to inform in the introduction the definition of MIP, that it is with dry needling and what is the difference from the conventional dry needling technique (he will only be sure in the text until line 156 and 367).

I´m not sure if a limitation of the study was due to the position adopted to assess the thigh, because during the needling it is not being able to assess the vasomotor response of the contralateral limb or other parts of the body to see if there was a reflex dilatation, since pain involves central and systemic mechanisms (central sensitization (CS)). Although you have cited the pre-stimulation phase – to check the thermal symmetry between both sides, not during or after the procedure. Maybe it is better to explain this in the discussion.

Rewrite, it is unclear whether group 2 is latent TrP and whether group 1 is active TrP: twenty chronic TrPs negative sciatica patients (n=20) participated in the study (118). Upon reading, the phrase in English was dubious whether it is negative for TrP or for sciatica. In line 250 and 261 you wrote more clearly.

I didn´t see if it was commented on item 2.1 how patients with sciatica due to piriformis syndrome were excluded. Nor if they were associated or not with fibromyalgia syndrome, varices, lesions in the skin, hip cellulitis (gynoid lipodystrophy).

On line 139, would not be 99.1% (347 thermograms)?

Line 155: (a sign of neuropathic pain) – it could also be other clinical situations as joint and other muscles referred pain (sacroiliac, quadratus lumborum) or also vascular ischemia.

Line 156:  two trigger points/tender areas – I didn't understand if it is limited to only 2 or more TrPs? Why not one? Travel´s criteria are clear for one TrP. The term “tender areas” was somewhat confused, as tender points/areas are related to fibromyalgia and do not generate referred pain. Maybe proofread the text.  

Maybe it is interesting to better explain the meaning of Exp(β) results  in a multinomial logistic regression model in this study.

DISCUSSION

What was the diagnosis of this TrPs negative sciatica subjects? It's unclear to the reader when you say they presented “weak” or deeply located, not palpable TrPs, which were overlooked. So the TrPs negative could be confirmed with ultrasound exam or inclusion of a table showing the definitive diagnosis of this negative group.  

You compared the positive TrPs group with what exactly? Perhaps a normal control group would be interesting to check if the thermal behavior is equal to the negative TrP group. This can be mentioned in the discussion as a limitation of the study.

I didn't understand in line 376: clinically important significance of ΔTavr. Why are 30 seconds not clinically important significance of ΔTavr? Which were the clinical criteria for this affirmation?

In line 384, you said “It seems possible to shorten the noxious stimulation phase.” However, the results for me are clear that the noxious stimulation could be much more shortened, as 30 seconds or 4 min. And it will be better for the clinic approach.

For me, it doesn't seem logical to say that the stimulus was toxic in the positive TrP, as it increased the temperature. And yes, it was toxic only in negative TrP, which decreased temperature. This was one of the most important findings of your study. I suggest you have a follow-up stating whether sciatica has improved in the vasodilation group, i.e., the clinical outcomes, compared to the non-vasodilation sciatica group.

CONCLUSION

I suggest emphasizing the better practice result of a significant ΔTavr increase (0.27 ± 0.26??) first confirmed at 30 s and a significant ΔTavr increase in above 0.5ºC confirmed for the TrPs-positive patients at 4’30’’ of the procedure. That is different from the abstract that there is no mention in the conclusion: “A continued significance of both parameters was confirmed from 6’00’’ to …….. vs TrPs(-) 3.77 ± 9.14%; p=0.000; CI (0.347,0.348)).”  Review the conclusion in the abstract.

Author Response

Dear Reviewer,

Thank you very much for the opportunity to further with the paper. According to your suggestion, we have carefully considered all opinions. Those comments helped us greatly to clarify the method. Please find our detailed response below:

INTRODUCTION: Despite being excellent, you should correct parts unclear in the introduction. Correcting the sentence in line 42 to 44. ST is also a quantitative and therefore objective method. As explained in the next sentence, however, correct them to greater than 0.3oC.

According to the reviewer’s suggestions, the temperature ranges have been changed from 0.3°C to 0.8°C.

ST has not raised many doubts nor considered to be of limited use, there is much more literature on the reliability of ST than ADT. In ST, there is also ANS stimulation when exposing the patient to a slightly cold controlled environment at 23oC. It is the room temperature cooler than the skin that will ensure better contrast and after colors and temperature adjustments in the image (line 49) – Despite being a retrospective analysis I neither didn´t find room temperature in the method nor if the patients did a period of thermalization inside the room before MIP. And analysis accuracy depends on the instrument's calibration and good reproducibility technique (line 49).

It was not our intention to disqualify ST as a reliable IRT method. We should be more precise and say that it has been criticized as a source of valuable diagnostic parameters by scientists from other fields. Because the ST method is not the main topic of our study and the paragraph on its reliability is not necessary, we decided to remove that controversial and inaccurate part (line 49).

As regards the IRT methodology, all terms indicated by the reviewer are available in the general protocol of the validation study quoted within the method section of the current paper.

VALIDATION STUDY Statement. Thermographic images were recorded by an expert following a standard protocol recommended by the Academy of Neuromuscular Thermography. The expert also evaluated the images. Patients were instructed to avoid physiotherapy and manual therapy 24 hours prior to the test and to avoid using nasal decongestants, analgesics, anti-inflammatory drugs, or any substances affecting the sympathetic function. They were also instructed not to drink coffee or alcohol and to refrain from smoking 2 hours before the recording.

To obtain the stability of the patient’s body temperature and to ensure the adjustment of the recording camera’s temperature to the interior conditions, the evaluation began 30 minutes after the patient had entered the examination room. Thermal isolation of the evaluated area from other thermal factors that might have influenced the evaluation, including other parts of the patient’s and doctor’s bodies, was ensured. Moreover, when performing thermovision imaging, the general rules of camera usage were followed. [ Skorupska E, Rychlik M, Samborski W. Validation and Test-Retest Reliability of New Thermographic Technique Called Thermovision Technique of Dry Needling for Gluteus Minimus Trigger Points in Sciatica Subjects and TrPs-Negative Healthy Volunteers. Biomed Res Int. 2015;2015:546497. doi: 10.1155/2015/546497. Epub 2015 Jun 7. PMID: 26137486; PMCID: PMC4475557.]

The current paper focused on the development of the protocol including Matlab, which demanded a very precise method description. We demonstrated a shortened and simplified version of the MIP description and presented a picture to enable an easier understanding to the reader. We feel that a more detailed explanation of some IRT measurement terms should be followed by even more detailed information, which would make the paper very long and complicated. 

I liked the term "autonomic referred pain (AURP)" but I do not agree in the introduction that it is only "a high temperature mark on the pain region" (line 80), there also be a decrease in temperature at the referred pain site, especially when provoked by noxious stimulus (painful compression of trigger points). So I understand that AURP has been a specific term for a temperature increase used specifically IN THIS STUDY, it cannot be generalized to the concept that there is always a temperature increase in the referred pain region. In line 152 you correctly used the expression vasodilatation or amplified VASOCONSTRICTION. Also, line 221.

Thank you very much for your attentive remark. We have corrected the sentences by adding a high or low temperature mark on the pain region (line 80)

METHOD

MATLAB® is a registered trademark. It should inform that it used a programming and numerical computing platform (MATLAB, MATrix LABoratory, version R2021, year).

The information on Matlab has been included in the statistical analysis section (line 256). Moreover, we have moved the software description up to line 140, where Matlab was mentioned in the Method section for the first time.

Describe what MIP is. The reader won't know it is needling if they don't read your previous papers. It is essential to inform in the introduction the definition of MIP, that it is with dry needling and what is the difference from the conventional dry needling technique (he will only be sure in the text until line 156 and 367).

The fast-in fast-out technique is widely used and considered a standard technique for TrPs. The static needle insertion to the point is characteristic of acupuncture. However, it is also used for TrPs.

The MIP description in lines 65-66 has been extended:

The nociceptive noxious stimulus was the fast-in fast-out dry needling technique.

Additionally, we have corrected the sentence in line 157 by adding: the noxious stimulation phase – 10 minutes of the fast-in fast-out dry needling of two trigger points (…).

I´m not sure if a limitation of the study was due to the position adopted to assess the thigh, because during the needling it is not being able to assess the vasomotor response of the contralateral limb or other parts of the body to see if there was a reflex dilatation, since pain involves central and systemic mechanisms (central sensitization (CS)). Although you have cited the pre-stimulation phase – to check the thermal symmetry between both sides, not during or after the procedure. Maybe it is better to explain this in the discussion.

The MIP is considered to be a completely new type of IRT. We did not consider the side-to-side comparison, which is an IRT standard. The patient's position during the MIP was established to observe possible referred pain due to noxious stimulation on the same side. Thus, the MIP is focused on a specific ROI with the expected vasomotor response confirmed in time and visible on highly contrastive thermograms. Thus, the patient position cannot be a limitation of the study. We confirmed the central sensitization process based on the biological background. TrPs are related to central sensitization, which is provoked and/or sustained mainly by the immune and autonomic nervous systems. Thus, intensive provoked vasodilatation explained by pathological autonomic activity is proof of the CS process for trigger points. This is the biological basis of our method, which is considered to confirm trigger points only.

All these things are explained in the validation study, and the current work is the continuation as it adds the Matlab automation to our method to improve its quality and reliability.

Rewrite, it is unclear whether group 2 is latent TrP and whether group 1 is active TrP: twenty chronic TrPs negative sciatica patients (n=20) participated in the study (118). Upon reading, the phrase in English was dubious whether it is negative for TrP or for sciatica. In line 250 and 261 you wrote more clearly.

Group 2 has no trigger points, neither active nor latent, according to Travell and Simons’ criteria.

We have changed the misleading TrPs (-) notation in the flow chart to TrPs negative.

I didn´t see if it was commented on item 2.1 how patients with sciatica due to piriformis syndrome were excluded. Nor if they were associated or not with fibromyalgia syndrome, varices, lesions in the skin, hip cellulitis (gynoid lipodystrophy).

Fibromyalgia syndrome is more complex pain. The inclusion criteria state, among others, unilateral sciatic pain being the dominant problem, which excludes fibromyalgia patients. Moreover, piriformis syndrome is due to nerve entrapment, and the pain starts from the buttock, not the lumbar side typical of sciatica.

On line 139, would not be 99.1% (347 thermograms)?

Unfortunately, we miswrote the number of MIP thermograms. The procedure lasted 16 minutes and the IRT pictures were recorded every 3 seconds. So, it means that 320 thermograms were taken. We corrected this mistake in the full paper.

Line 155: (a sign of neuropathic pain) – it could also be other clinical situations as joint and other muscles referred pain (sacroiliac, quadratus lumborum) or also vascular ischemia.

We cannot agree that thermal asymmetry is the sign of the muscle referred pain. The literature indicates that noxiously-provoked TrPs produce attenuated vasoconstriction or the IRT is not sensitive enough to detect changes. This forms the basis of the MIP’s unusual reactivity and its diagnostic value.

The QL produces sacro-lumbar pain – not sciatica, etc.

Line 156:  two trigger points/tender areas – I didn't understand if it is limited to only 2 or more TrPs? Why not one? Travel´s criteria are clear for one TrP. The term “tender areas” was somewhat confused, as tender points/areas are related to fibromyalgia and do not generate referred pain. Maybe proofread the text. 

To keep comparable methodology conditions, we considered the noxious stimulation of two areas. If the patients presented myofascial pain, we aimed at the two most sensitive TrPs. For TrPs negative patients, we considered the two most sensitive areas. However, we agree that because the term tender point is strongly associated with fibromyalgia, the sentence needs rephrasing to:

(2) the noxious stimulation phase – 10 minutes of fast–in fast-out dry needling of two trigger points (for MPS patients)tender  or the two most sensitive areas (for TrPs negative patients) within a muscle and the IRT observation of the region assumed to be the referred pain zone of the examined trigger point/muscle.

Maybe it is interesting to better explain the meaning of Exp(β) results  in a multinomial logistic regression model in this study.

This is a statistical model used to analyze all MIP parameters at once (Tavr and AURP segmented picture with Tmax and Tmin being the oriented point to observe the isothermal area defined as a vasodilatation or vasoconstriction mark). It may not be common, but it is an advanced statistical method of multiple data analysis and it is used to indicate which parameter is the most important to detect changes significantly.

If the Exp(β) is lower than 1, the expected value of the examined variable decreases and vice versa – if the Exp(β) is greater than 1, the expected value of the examined variable increases. So, on the one hand, the Exp(β) of 0.8 means that as the independent variables increase, the chance to obtain higher values for the dependent variable is getting LOWER by 20%. On the other hand, the Exp(β) of 1.5 means that as the independent variables increase, the chance to obtain higher values for the dependent variable is getting HIGHER by 50%.

DISCUSSION

What was the diagnosis of this TrPs negative sciatica subjects? It's unclear to the reader when you say they presented “weak” or deeply located, not palpable TrPs, which were overlooked. So the TrPs negative could be confirmed with ultrasound exam or inclusion of a table showing the definitive diagnosis of this negative group. 

There is no one objective tool to confirm TrPs. The USG is at the experimental phase and was mainly used on the superficial trapezius. All of the studies in the trigger points field are based on the palpatory criteria established by Travell and Simons. We add the referred pain criterion to strengthen the methodology of the study.

As regards your doubts: Our consideration was that the most deeply located TrPs of the gluteus minimus muscle lying 5 cm down under the skin cannot be detected by snapping palpation during the diagnostic procedure (Travell&Simons) but they can be easily reached by needle encounter. Thus, in the palpated protocol the patients are TrPs negative, but in our opinion dry needling muscle irritation can reach those “hidden TrPs”. This can be an advantage of the MIP method, which should be developed and explained by conducting further studies. For your information, we have performed such an analysis and the results indicated that it is possible to reach some “hidden TrPs” (the paper under review).

You compared the positive TrPs group with what exactly? Perhaps a normal control group would be interesting to check if the thermal behavior is equal to the negative TrP group. This can be mentioned in the discussion as a limitation of the study.

We agree that it is a weakness of the paper, but it was a retrospective study focused on the Matlab implementation into the MIP data analysis. We agree that the method should be further developed. We believe that using Matlab will enable and facilitate this development in the future.  

I didn't understand in line 376: clinically important significance of ΔTavr. Why are 30 seconds not clinically important significance of ΔTavr? Which were the clinical criteria for this affirmation?

The MIP is a completely new type of ADT, where we provoke amplified reactions in the area of the theoretically established referred pain pattern of trigger points. The static thermography side-to-side comparison cannot serve as the baseline for the MIP. The MIP is considered to provoke a diagnostic phenomenon approved by a series of highly contrastive pictures (320 thermograms).

Please note that one thermogram’s significance can be perceived as artificial. The MIP showed a gap in the significance of the ΔTavr increase between 30” and 4’30”. From the point of 4’30”, a continued significance of the ΔTavr increase was observed. We stated the level of 0.5°C as clinically important based on the musculoskeletal IRT literature indicating various temperature as indicative of disease confirmation, e.g. 0.3°C or 0.5°C to 0.7°C or 0.8°C. We agree that future studies based on the MIP results will allow us to find out what is the temperature increase indicative of TrPs in general. However, this will demand a bigger group of patients with/without TrPs, control, etc.

In line 384, you said “It seems possible to shorten the noxious stimulation phase.” However, the results for me are clear that the noxious stimulation could be much more shortened, as 30 seconds or 4 min. And it will be better for the clinic approach.

We cannot agree with the reviewer’s opinion. To confirm TrPs, the MIP demands a simultaneous development of its both parameters (Tavr and AURP), which was reached at 3’30”. Moreover, a continued significance of both parameters was confirmed from 6’00’’ to 15’30’’ of the procedure. Thus, based on this study the stimulation phase could be stopped at 6’00”, but we do not know if it is characteristic of that group only or of all patients in general. We should check that in the future.

Importantly, the MIP can confirm a disease if both diagnostic parameters are significant in a high percentage of the recorded IRT pictures. The main idea behind the MIP is to dispute any possible doubts that the observed phenomenon is just artificial.

For me, it doesn't seem logical to say that the stimulus was toxic in the positive TrP, as it increased the temperature. And yes, it was toxic only in negative TrP, which decreased temperature. This was one of the most important findings of your study. I suggest you have a follow-up stating whether sciatica has improved in the vasodilation group, i.e., the clinical outcomes, compared to the non-vasodilation sciatica group.

Thank you kindly for the suggestion about the TrPs negative sciatica group. However, the MIP is meant to objectively diagnose referred pain due to central sensitization. Firstly, we should establish and prove the full MIP protocol that would include all the terms important for ADT methods, e.g. defining the recovery phase or establishing the stimulation phase duration. Only then can we think about the clinical method utility, which is very interesting for us as well. We understand that the reviewer was thinking about the MIP results utility for the clinical practice. This is our aim in the future.

CONCLUSION

I suggest emphasizing the better practice result of a significant ΔTavr increase (0.27 ± 0.26??) first confirmed at 30 s and a significant ΔTavr increase in above 0.5ºC confirmed for the TrPs-positive patients at 4’30’’ of the procedure. That is different from the abstract that there is no mention in the conclusion: “A continued significance of both parameters was confirmed from 6’00’’ to …….. vs TrPs(-) 3.77 ± 9.14%; p=0.000; CI (0.347,0.348)).”  Review the conclusion in the abstract.

This is the first paper presenting the Matlab implementation into the MIP. The automation allowed us to observe and confirm the observed phenomenon more precisely. Thus, as we followed the results, we first described how a single parameter reacted in time, and then we indicated that the MIP demands a simultaneous increase in both parameters (ΔTavr and AURP). This was obtained from 6’00’’ to 15’30’’ as underlined in the abstract.

However, we would like to thank the reviewer for the comment pointing out how to simplify the MIP description. Our intention was to provide as detailed explanation of the parameters as possible to stop any doubts. Thus, we are the main critics of our method, which can become one of the first objective tools for some subtypes of the central sensitization process.

To underline the MIP terms, we have added to the paper: For an objective TrPs confirmation, the MIP demands a simultaneous increase in both parameters.

Reviewer 2 Report

The article describes a novel and interesting method to measure autonomic phenomenon associated with trigger points. The manuscript is written correctly and presents a detailed technical description on the software data analysis. However, there are few aspects that I believe could be improved in the manuscript. Please find them below:

  • Data regarding the age of the sample and the number of subjects included in each group should be included at the beginning of the results (I think this it is more recommended than in the method section)
  • Could you please describe in the manuscript the method used in order to palpate gluteus minimus muscle to confirm the spot tenderness for TrPs diagnosis?. It may be a challenging diagnosis considering that the gluteus medius is more superficial, so the tender spots palpated on surface could be from gluteus medius. This will also affect the pain recognition criteria when compressing the TrP.
  • Was the TrP diagnostic criteria of limited range of movement standardized in any way? Which test was used in order to conclude that it is considered short?
  • I miss some more information about the process of the noxious stimulation phase. Please describe the procedure of dry needling in the manuscript (Whether the needle was moved or kept static, if LTRs were elicited, etc).
  • Was the therapist who applied dry needling blinded to the subject group allocation?. Please specify in the manuscript. It can be a potential source of bias.
  • As the manuscript describes the automatization of a new thermography method, has been the reliability of this method investigated in previous research? Or could you provide any reliability data from this investigation?
  • The authors should include the descriptive and clinical data of the population at the beginning of the results section. I cannot find any data for each group such as age, BMI, duration of symptoms, intensity of sciatic pain, etc. In addition, the authors should include an statistical analysis to compare these values at baseline, as it will allow to confirm that both groups characteristics were similar (E.g: there were not differences between group on age or pain intensity).
  • I cannot clearly find a limitation section or a paragraph detailing the limitations of the study at the end of the discussion
  • It will be interesting to include what are the clinical implications of the study in the discussion section.

Author Response

Dear Reviewer,

Thank you very much for the opportunity to further with the paper. According to your suggestion, we have carefully considered all opinions. Those comments helped us greatly to clarify the method. Please find our detailed response below:

  • Data regarding the age of the sample and the number of subjects included in each group should be included at the beginning of the results (I think this it is more recommended than in the method section)

Thank you very much for your opinion. However, because the study is focused on automated thermogram analysis, age and gender are not important for the study results analysis. Moreover, this type of data can be put either in the Method or the Results section. Additionally, you suggested adding the baseline characteristics of study participants. We have added participants' characteristics to the Method section and we hope that our changes will satisfy you.

There was no significant difference between TrPs positive and TrPs negative sciatica patients. The demographic data were respectively as follows: (i) age 44 ± 6.6 vs 46 ± 7.5; (ii) pain duration 11.3 ± 7.7 vs 10.7 ± 7.5; (iii) VAS 6.17 ± 2.1 vs 5.35 ± 1.6; (iv) BMI 23.29 ± 3.1 vs 26.49 ± 4.1;

  • Could you please describe in the manuscript the method used in order to palpate gluteus minimus muscle to confirm the spot tenderness for TrPs diagnosis?. It may be a challenging diagnosis considering that the gluteus medius is more superficial, so the tender spots palpated on surface could be from gluteus medius. This will also affect the pain recognition criteria when compressing the TrP.

Please look at the method description. We stated that the taut band of gluteus minimus TrPs is not accessible in most cases. Thus, we added the referred pain pattern characteristic of gluteus minimus as a diagnostic criterion. This pattern is completely different from the one for the overlying gluteus medius. This ensures that there is no doubt as to which TrPs were irritated. Pain recognition was focused on sciatic pain indicated for gluteus minimus. The pattern of gluteus medius is limited to the sacral region only. In our opinion, all this makes a misdiagnosis impossible.

  • Was the TrP diagnostic criteria of limited range of movement standardized in any way? Which test was used in order to conclude that it is considered short?

TrPs provoke pain related ROM limitation. There is no specific test that could be used to confirm that. We just checked if the hip abduction and flexion were restricted according to the standard ROM.

  • I miss some more information about the process of the noxious stimulation phase. Please describe the procedure of dry needling in the manuscript (Whether the needle was moved or kept static, if LTRs were elicited, etc).

The type of dry needling has been specified in the manuscript (fast-in fast-out). It is not possible to observe LTRs for the examined muscle. The only possible confirmatory sign is the referred pain, which we added as a diagnostic criterion. Additionally, the MIP confirmed its presence objectively.

  • Was the therapist who applied dry needling blinded to the subject group allocation?. Please specify in the manuscript. It can be a potential source of bias.

The therapist who performed the MIP noxious stimulation was not blinded to the TrPs diagnosis results. It can be a potential source of bias, especially in clinical studies.

  • As the manuscript describes the automatization of a new thermography method, has been the reliability of this method investigated in previous research? Or could you provide any reliability data from this investigation?

The method has been examined for both validity and reliability (test re-test). We write about it in line 151.

However, as a result of your comment, we have added a more precise description to this sentence:

According to the validation study (both validity and reliability (test re-test), two parameters are indicative of TrPs presence: ΔTavr and AURP The results of the Matlab analysis were also cross-checked with order statistics packages (R packages) and all the calculations were replicated.

  • The authors should include the descriptive and clinical data of the population at the beginning of the results section. I cannot find any data for each group such as age, BMI, duration of symptoms, intensity of sciatic pain, etc. In addition, the authors should include an statistical analysis to compare these values at baseline, as it will allow to confirm that both groups characteristics were similar (E.g: there were not differences between group on age or pain intensity).

All the data you requested have been added to the Methods section.

  • I cannot clearly find a limitation section or a paragraph detailing the limitations of the study at the end of the discussion

We have added a limitation section according to your suggestion

  • It will be interesting to include what are the clinical implications of the study in the discussion section.

To the end of the discussion we add:

The clinical implications of the study

The MIP is intended to objectively confirm the trigger points presence. Thus, it gives an opportunity to stop controversies around their presence. However, further studies considering other muscles are necessary.

Kind regards,

Authors
